# RAGGED: Towards Informed Design of Scalable and Stable RAG Systems

**Jennifer Hsia** [* 1]   **Afreen Shaikh** [* 2]   **Zhiruo Wang** [2]   **Graham Neubig** [2]

## Abstract

Retrieval-augmented generation (RAG) enhances language models by integrating external knowledge, but its effectiveness is highly dependent on system configuration. Improper retrieval settings can degrade performance, making RAG less reliable than closed-book generation. In this work, we introduce RAGGED, a framework for systematically evaluating RAG systems across diverse retriever-reader configurations, retrieval depths, and datasets. Our analysis reveals that reader robustness to noise is the key determinant of RAG stability and scalability. Some readers benefit from increased retrieval depth, while others degrade due to their sensitivity to distracting content. Through large-scale experiments on open-domain, multi-hop, and specialized-domain datasets, we show that retrievers, rerankers, and prompts influence performance but do not fundamentally alter these reader-driven trends. By providing a principled framework and new metrics to assess RAG stability and scalability, RAGGED enables systematic evaluation of retrieval-augmented generation systems, guiding future research on optimizing retrieval depth and model robustness. [1]

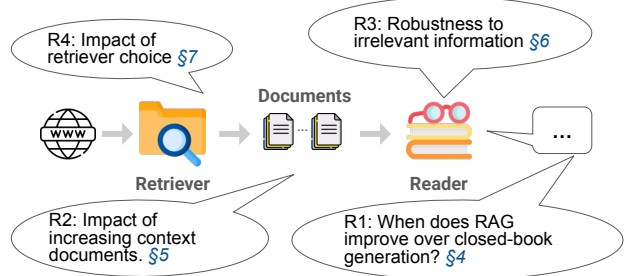

Figure 1: Roadmap of what our framework RAGGED analyses across the RAG pipeline.

## 1. Introduction

Retrieval-augmented generation (RAG) (Chen et al., 2017; Lewis et al., 2020) enhances large language models (LLMs) by retrieving relevant external contexts, enabling more specific and factually grounded responses. However, despite its promise, RAG's effectiveness is not guaranteed. In fact, improper configurations can degrade model performance, leading to outputs that are worse than closed-book genera-

tion. Understanding when and why RAG helps or harms is critical for optimizing system design.

Most prior work evaluates RAG under controlled conditions and curated contexts (Liu et al., 2023; Cuconasu et al., 2024), which fail to reflect real-world retrieval challenges. In practice, retrieved contexts contain both relevant and irrelevant information, making the reader model's ability to filter noise a critical factor in RAG success. Additionally, prior studies provide conflicting findings on retrieval depth ($k$)— while some suggest increasing $k$ improves performance (Izacard & Grave, 2021), others observe diminishing returns (Liu et al., 2023) or even degradation at high $k$ (Cuconasu et al., 2024; Jiang et al., 2024). This lack of consensus leaves practitioners without clear guidance on how to configure RAG systems for different tasks.

To address these challenges, we introduce **RAGGED (Retrieval-Augmented Generation Generalized Evaluation Device)**, a framework for systematically evaluating RAG performance across retrieval depths, model architectures, and retrieval conditions. Unlike prior work, which often relies on synthetic or manual retrieval modifications, RAGGED assesses models under realistic retrieval scenarios — analyzing performance on naturally retrieved top-$k$ contexts rather than manually curated, oracle-aware contexts.

Our study reveals that reader robustness to noise is the primary factor driving RAG stability and scalability, rather than retriever quality alone. To quantify this, we introduce two new metrics: the **RAG Stability Score (RSS)** and **RAG Scalability Coefficient (RSC)**, providing a princi-

*Equal contribution [1]Machine Learning Department, Carnegie Mellon University, Pittsburgh, USA [2]The Language Technologies Institute, Carnegie Mellon University, Pittsburgh, USA. Correspondence to: Jennifer Hsia <jhsia2@andrew.cmu.edu>.

[1]Code and data for the RAGGED framework are available at https://github.com/neulab/ragged

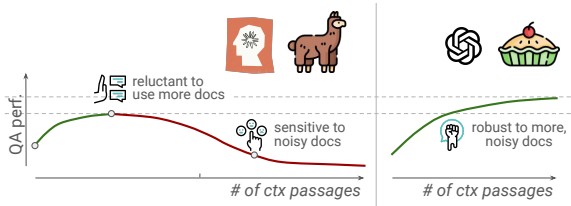

Figure 2: While some readers exhibit 'peak-then-decline' (left), others exhibit 'improve-then-plateau' behavior (right) with increasing number of contexts.

pled framework for evaluating retrieval effectiveness across diverse configurations.

Using RAGGED, we conduct a large-scale empirical study to answer four key questions (Figure 1), each corresponding to a core section of our paper:

1. **Under What Conditions Does Retrieval Outperform Closed-Book Generation?** (§4) We analyze when retrieval improves performance and identify that some readers frequently benefit from RAG, particularly at large $k$, while others degrade due to noise sensitivity.

2. **How Does Retrieval Depth Impact Stability and Scalability?** (§5) We identify two distinct reader behaviors: improve-then-plateau models, which scale effectively, and peak-then-decline models, which degrade at higher $k$ (Figure 2).

3. **How Do Readers Handle Noisy Retrieval, and Is Prompting a Reliable Fix?** (§6) We evaluate RAG performance under realistic retrieval conditions, showing that noise sensitivity—rather than retriever quality alone—determines downstream effectiveness. We also assess whether instructing readers to focus on relevant content mitigates noise sensitivity.

4. **When Does a Better Retriever Actually Lead to Better Performance?** (§7) While retriever choice shifts overall performance, it does not alter fundamental reader behaviors, thus highlighting the reader as the key driver of stability and scalability.

By introducing a structured and reproducible evaluation framework, our study provides foundational insights into the dynamics of RAG systems and guides future research toward optimizing retrieval-augmented generation for real-world applications.

## 2. The RAGGED Framework

Evaluating retrieval-augmented generation (RAG) remains challenging due to inconsistencies across retrieval depths,

datasets, and reader models. Prior evaluations often rely on oracle-aware curation of contexts or fixed retrieval configurations, thereby failing to capture how RAG systems behave under real-world retrieval conditions. These limitations obscure the key factors that determine RAG effectiveness, particularly in terms of **stability** (consistency near the retrieval depth that yields optimal performance) and **scalability** (sustained performance gains as retrieval depth increases).

To address these gaps, we introduce RAGGED, a systematic framework for evaluating RAG systems across diverse retrieval settings. RAGGED provides a principled approach to assessing model behavior and optimizing retrieval depth for robust performance.

**Objectives of RAGGED:** RAGGED provides a structured evaluation of RAG effectiveness, enabling:

- **Retrieval-depth analysis**: Identifying whether increasing $k$ improves or harms model performance.

- **Stability assessment**: Measuring how consistently models maintain performance near their optimal retrieval depth, avoiding sharp performance drops.

- **Scalability evaluation**: Determining whether a model continues to benefit from increasing retrieval depth or experiences diminishing returns.

- **Reproducible benchmarking**: Standardizing evaluation across different RAG implementations.

To operationalize these goals, we introduce two new metrics:

**RAG Stability Score (RSS)** The RSS metric quantifies how consistently a model maintains performance around its optimal retrieval depth ($k^*$). A stable model should exhibit minimal performance fluctuation within a small retrieval window, while an unstable model's performance will degrade noticeably when retrieval depth varies slightly. RSS is formally defined as:

$$RSS = \frac{\min\limits_{k \in [k^*-\Delta, k^*+\Delta]\setminus\{k^*\}} \text{Performance at } k}{\text{Performance at } k^*} \quad (1)$$

where $k^*$ is the retrieval depth that yields peak model performance, and $\Delta$ defines a local window (e.g., $k^* \pm 5$) to assess stability. The numerator captures the minimum performance in this range, excluding $k^*$, effectively measuring the worst-case volatility.

A higher RSS ($\approx 1.0$) indicates that performance remains steady across nearby retrieval depths, suggesting robustness to retrieval variations. A lower RSS ($RSS \ll 1.0$) signals sharp fluctuations near $k^*$, implying sensitivity to retrieval depth choices and reduced stability.

While RSS currently uses a symmetric window to evaluate performance variation around the optimal retrieval depth, we acknowledge that retrieval effects can be directionally asymmetric – adding irrelevant context (right side) may degrade performance differently than omitting high-quality content (left side). However, our empirical analysis shows that this asymmetry is not consistent across models (e.g., LLAMA2 vs. LLAMA3), thus supporting the use of a symmetric window as a general-purpose diagnostic. That said, directional extensions of RSS could offer deeper insight into model fragility under under- or over-retrieval, which we leave to future work.

**RAG Scalability Coefficient (RSC)** The RSC metric captures the total accumulated benefit a model gains as retrieval depth increases before performance plateaus or declines. A model with high scalability should continue to improve with additional retrieved contexts, while a low-scalability model will either plateau early or exhibit only minimal improvement. The RSC is defined as:

$$RSC = \sum_{i=1}^{i_{\text{last gain}}} \frac{1}{2}(k_{i+1} - k_i)(\text{F}_{1i} + \text{F}_{1i+1}) \qquad (2)$$

where $k_{\text{last gain}}$ is the last retrieval depth before performance plateaus or declines. This is determined as the last $k$ where:

$$\text{F}_{1k} - \text{F}_{1k-1} \geq \epsilon$$

for a predefined threshold $\epsilon$.

A higher RSC reflects sustained improvements over a broader range of retrieval depths, demonstrating strong scalability. Conversely, a lower RSC suggests the model stops improving early, indicating limited scalability.

By providing a structured evaluation across diverse retriever-reader configurations, RAGGED enables systematic comparisons of different RAG systems and offers insights into optimizing retrieval depth for robust performance.

In particular, the metrics RSS and RSC serve as complementary diagnostics rather than substitutes for task-specific metrics like $\text{F}_1$: RSS quantifies how brittle or robust a model is to changes in retrieval depth near its peak performance, while RSC assesses how much a model continues to benefit from additional retrieval. A model may achieve high performance at a single retrieval depth but still have low RSS (unstable across depths) or low RSC (unable to scale with more information). Conversely, a model with slightly lower peak performance but high RSS/RSC may be more robust and easier to deploy in real-world settings where retrieval conditions fluctuate.

Together with task-specific performance scores, these diagnostic metrics provide a more complete picture of model behavior. They help developers understand not only how well a system performs, but also how reliably and efficiently it does so across changing retrieval conditions.

**Hyperparameter Justification and Metric Stability** We set $\epsilon$ for RSC to the 70th percentile of all positive $\Delta\text{F}_1$ values across models and retrieval depths, and $k = 5$ for RSS. This percentile rule balances sensitivity to meaningful gains with robustness against small fluctuations, ensuring discriminative power across models. We verified that model rankings remain unchanged when varying $\epsilon$ between the 65th and 75th percentile and $\delta$ from $\pm 5$ to $\pm 10$, indicating that both RSS and RSC are robust to reasonable parameter shifts and provide consistent comparative signals across models.

## 3. Experimental Setup

We implement the RAGGED framework by evaluating retrievers and readers across multiple retrieval depths, analyzing how models respond to increasing context sizes and retrieval noise.

### 3.1. Retrievers

We evaluate three retrievers with different retrieval paradigms: (1) BM25 (Robertson et al., 2009), a sparse lexical retriever based on term matching. (2) ColBERT (Santhanam et al., 2021), a neural retriever using contextualized late interaction. (3) Contriever (Izacard et al., 2022), an unsupervised dense retriever emphasizing document-level semantic similarity.

### 3.2. Readers

We analyze both closed-source and open-source reader models: **Open-source:** We use FLANT5-XXL (Chung et al., 2022) (11B parameters) and FLAN-UL2 (Tay et al., 2023) (20B), as well as LLAMA2 (Touvron et al., 2023) (7B, 70B) and LLAMA3 (8B, 70B).

**Closed-source:** We evaluate GPT-3.5-turbo (16k context length) and CLAUDE-3-HAIKU (200k). A subset of experiments includes GPT-4O with a 128k context window. We include both open- and closed-source models to ensure that our findings generalize across different architectures and training paradigms.

### 3.3. Datasets

We evaluate RAG performance across three datasets spanning different reasoning complexities and domain-specificity (Table 2, Table 3):

- **Natural Questions (NQ)** (Kwiatkowski et al., 2019): Wikipedia-based, single-hop QA with real user queries.

- **HotpotQA** (Yang et al., 2018): Wikipedia-based, multi-hop QA requiring reasoning over multiple passages.

- **BioASQ (Task 11B)** (Krithara et al., 2023): PubMed-based biomedical QA for specialized domains.

These datasets allow us to assess RAG performance across general knowledge (NQ), complex reasoning (HotpotQA), and domain-specific retrieval (BioASQ).

### 3.4. Metrics

We evaluate both retrieval and reader performance following best practices from Petroni et al. (2021).

**Retriever Performance:** We report **recall@k**, which measures the fraction of ground-truth passages present in the top-$k$ retrieved results. Higher recall indicates better retrieval coverage but does not guarantee better reader performance.

**Reader Performance:** We compute **unigram $F_1$**, which measures lexical overlap between model predictions and gold answers. Each query is evaluated against all gold answers, and the highest score is reported. To further assess correctness, we validate key results using an LLM-based semantic correctness metric (Kim et al., 2024) on a subset of responses (Appendix J).

To analyze retrieval-depth stability and scalability, we evaluate the **RAG Stability Score (RSS)** and **RAG Scalability Coefficient (RSC)** as defined in section 2. These metrics provide a principled way to assess RAG systems beyond traditional retrieval and reader accuracy measures.

## 4. Under What Conditions Does Retrieval Outperform Closed-Book Generation?

Retrieval-augmented generation (RAG) is widely assumed to enhance model performance by providing external knowledge, but our findings reveal that its effectiveness is highly model-dependent. While some models benefit from retrieved context, others perform worse than when no retrieval is used at all. This section investigates when retrieval actually helps, which models are most affected by retrieval noise, and how retrieval effectiveness varies across tasks and domains.

### 4.1. When Does Retrieval Help?

Retrieval effectiveness is primarily determined by the model's ability to selectively use relevant information while ignoring misleading or redundant content. We observe two distinct behaviors:

First, **some models benefit consistently from retrieval**, showing significant performance improvements when retrieval is enabled. Models such as FLAN and GPT-3.5 con-

sistently achieve gain with RAG, suggesting that they can effectively extract useful information from retrieved passages while discarding irrelevant details. However, just because a reader consistently gains, does not mean the gain amount is significant. For example, across datasets, FLANT5 achieves an average gain of 16-30 $F_1$ points whereas GPT-3.5 achieves an average gain of 1 to 9 $F_1$ points in comparison to closed-book generation.

In contrast, **some models degrade with retrieval**, sometimes performing worse than their no-context baseline. Models like LLAMA and CLAUDE struggle with filtering noisy retrievals, which results in lower accuracy when using RAG. Instead of leveraging additional knowledge, these models become more susceptible to incorrect or distracting passages.

This suggests that retrieval is not inherently beneficial, but instead depends on how well a reader can balance the trade-off between extracting useful knowledge and avoiding retrieval noise.

While retrieval can provide additional signal (relevant knowledge), it also introduces noise (irrelevant passages). The models that benefit most from retrieval tend to be those that can effectively distinguish between high-value and low-value context, whereas noise-sensitive models treat all retrieved passages equally, leading to instability. We discuss some hypothesis Appendix E reader architecture and training details to trends.

### 4.2. Task-Specific Retrieval Trends

We observe that retrieval effectiveness is not uniform across tasks and domains.

**Multi-hop questions benefit more from retrieval than single-hop questions.** Since multi-hop reasoning requires synthesizing multiple pieces of information, and can not be tackled simply by retrieving a short fact learned from pretraining. Thus, retrieval can be particularly helpful for multi-hop settings.

### Key Takeaways

Our results show that retrieval is not inherently helpful. Its effectiveness depends on the model's ability to handle noisy information. While some models consistently benefit from retrieval, others degrade due to over-reliance on irrelevant or misleading content. This highlights the need for retrieval-aware reading mechanisms that allow models to selectively integrate useful passages rather than treating all retrieved content equally.

| Dataset | ColBERT | BM25 | Dataset | ColBERT | BM25 |
|---------|---------|------|---------|---------|------|
| | **GPT-3.5-turbo** | | | **Claude-3-haiku** | |
| NQ | only for $k \geq 5$ | ✗ | NQ | ✗ | ✗ |
| HotpotQA | ✓ | ✓ | HotpotQA | only for $k \leq 2$ | ✓ |
| BioASQ | ✓ | ✓ | BioASQ | ✗ | ✗ |
| | **FlanT5** | | | **FlanUL2** | |
| NQ | ✓ | ✓ | NQ | ✓ | only for $k > 3$ |
| HotpotQA | ✓ | ✓ | HotpotQA | ✓ | ✓ |
| BioASQ | ✓ | ✓ | BioASQ | ✓ | ✓ |
| | **Llama2 7B** | | | **Llama2 70B** | |
| NQ | only for $k < 10$ | ✓ | NQ | only for $k < 20$ | ✓ |
| HotpotQA | ✓ | ✓ | HotpotQA | ✓ | ✓ |
| BioASQ | ✓ | ✓ | BioASQ | only for $k < 20$ | only for $k < 20$ |
| | **Llama3 8B** | | | **Llama3 70B** | |
| NQ | only for $k = 2$ | ✗ | NQ | ✗ | ✗ |
| HotpotQA | only for $k \leq 5$ | only for $k \leq 5$ | HotpotQA | only for $k \leq 5$ | only for $k \leq 5$ |
| BioASQ | only for $k \leq 2$ | only for $k \leq 2$ | BioASQ | only for $k = 1$ | ✗ |

Table 1: ✓ means the particular reader-retriever combination performs better than closed-book generation for all $k$'s. On the other hand, ✗ signifies that the particular reader-retriever combination consistently performs worse than closed-book generation, regardless of $k$. Otherwise, we describe the $k$-condition for which the retriever-reader combination performs better than closed-book generation.

## 5. How Does Retrieval Depth Impact Stability and Scalability?

Prior work reports conflicting effects of increasing retrieval depth ($k$): some studies find that performance saturates at high $k$ (Liu et al., 2023), while others observe degradation (Cuconasu et al., 2024; Jiang et al., 2024). Although these findings appear contradictory, we argue that they are actually complementary, as each study focuses on a limited range of retrievers, readers, and datasets. Our experiments, which span a wider variety of retrievers, readers, and datasets, demonstrate that both saturation and degradation behaviors can occur with the determining factor being the choice of reader model.

Specifically, we observe two distinct trends in reader performance (Figure 3):

**Improve-then-Plateau Models** Models such as FLAN and GPT-3.5 improve as $k$ increases and plateau around $k = 10$. **For these models, increasing $k$ maximizes performance without significant risk of degradation.**

**Peak-then-Decline Models** In contrast, models like LLAMA and CLAUDE-3-HAIKU peak at small $k$ (around $k < 5$) but degrade as $k$ increases due to their sensitivity to retrieval noise. **For these models, a small $k$ is optimal to**

minimize performance drops.

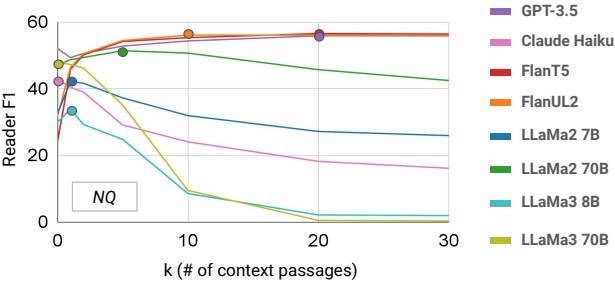

Figure 3: Reader performance on the NQ dataset as $k$, the number of contexts retrieved by ColBERT, varies. Colored circles indicate reader performance at the optimal $k^*$. Similar trends hold across retrievers (BM25, ColBERT, Contriever) and datasets (NQ, HotpotQA, BioASQ) in Figure 14 and Figure 15.

**Why This Matters** A model's response to increasing $k$ affects not only its peak performance but also its **stability** and **scalability**.

A well-designed RAG system should be **scalable**, meaning it should benefit from increasing $k$. Multi-hop reasoning tasks, in particular, require synthesizing multiple pieces of

information, making large $k$ essential. Peak-then-Decline models struggle in such cases.

We should also strive for **stable** model that maintains consistent performance near the optimal $k^*$, so that retrieval depth tuning is practical. Peak-then-decline models can exhibit a sharp performance drops even when $k$ is only 1 off from $k^*$, making them harder to tune and unreliable in practice.

To quantify these trends, we compute the RAG Scalability Coefficient (RSC) Figure 4 and the RAG Stability Score (RSS) Figure 5. We note that improve-then-plateau models have high RSC and RSS, which aligns with the intuition. Models like FLANT5 and GPT-3.5 exhibit high RSS scores, indicating strong performance stability across retrieval depths. In particular, FLANT5 achieves an RSS of 0.99, reflecting near-constant performance around its optimal k. We confirmed that this is not due to input truncation masking additional retrieved content, where we provide supporting analysis in Appendix G.

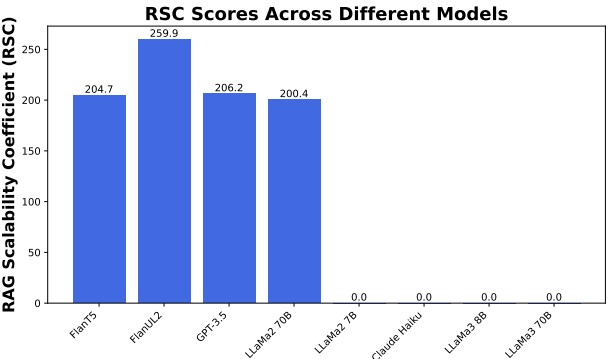

Figure 4: Ragged scalability coefficient for NQ, retriever colbert.

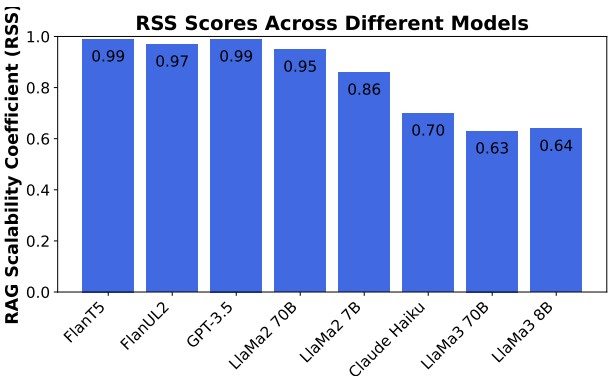

Figure 5: Ragged stability score for NQ, retriever colbert.

What is more interesting is that while a scalable model is often more stable, a stable model does not have to be a scalable one. For example, LLAMA2 models are not as scalable but are still stable. Although they are peak then decline models, they decline steadily. This reminds us that we really should aspire to optimize for both metrics, not just one.

To assess how well our findings generalize beyond traditional QA datasets, we conduct a preliminary evaluation on CRAG, a newer and more challenging RAG benchmark (Yang et al., 2024). We observe that the same reader-specific retrieval-depth trends hold: LLAMA2 exhibits early degradation, while FLANT5 remains stable across increasing k. Full results are provided in Appendix F.

**Key Takeaways** More retrieval is not always better—some models improve with increasing $k$, while others degrade due to noise sensitivity. More importantly, the kind of improvement future research should strive for is better scalability and stability. RAGGED provides a principled way to assess these two critical aspects, helping practitioners determine optimal retrieval depth per model.

## 6. How Do Readers Handle Noisy Retrieval, and Is Prompting a Reliable Fix?

Real-world RAG systems retrieve a mix of relevant and irrelevant content, making reader robustness to noise a key factor in performance. This section evaluates reader behavior when (1) at least one gold passage is present and (2) no gold passage is retrieved. The former setting represents a good scenario when there is sufficient signal to answer the question, the latter setting represents the worst-case scenario where there is not enough information to answer the question.

Throughout this section, we define noise as naturally retrieved, non-gold passages from actual retrievers. These are not artificially injected distractors but instead reflect real-world retrieval failures, such as topically related but misleading or irrelevant content.

### 6.1. With Gold Passages

We compare three conditions: (1) **Top-$k$**: full retrieved set, (2) **Top-gold**: only gold passages within the top-$k$, and (3) **No-context**: no retrieval. This evaluates how distracting noise compared to the signal (Figure 6).

**Reader Robustness Determines Gains from Retrieval** While robust models (e.g., FLAN, GPT) consistently improve with retrieval, noise-sensitive models (e.g., LLAMA, CLAUDE) degrade below their no-context baselines. This suggests that some models are not as good at performing the second-step filtering, leaving them vulnerable to noise when irrelevant passages are included. Future work should explore fine-tuning readers on diverse, noisy retrieval settings to improve robustness.

**Multi-hop Questions Mitigate Noise Effects** In Hot-

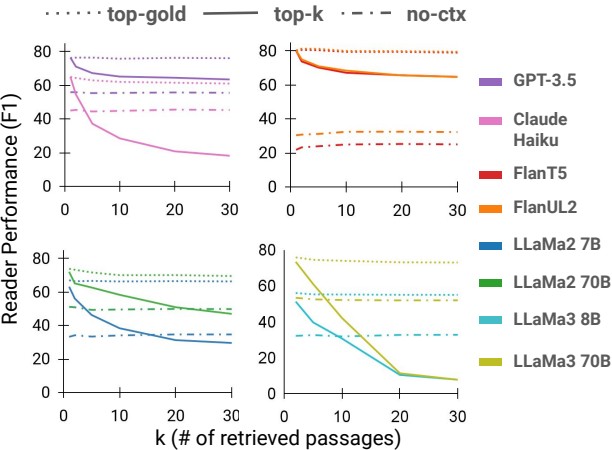

Figure 6: NQ results when at least one gold passage is in the top-$k$. 'Top-gold' includes only gold passages.

potQA, models maintain accuracy above no-context longer than in NQ. We hypothesize that multi-hop signals force the model to rely on more signal anchors, reducing reliance on single-passage heuristics and making models more resilient to noise.

**Domain-Specific Jargon Strengthens Retrieval** In BioASQ, the gap between top-gold and top-$k$ is smaller than in open-domain datasets, indicating that domain-specific terminology provides stronger retrieval cues. However, noise-sensitive models (e.g., CLAUDE-3-HAIKU and LLAMA3) still fall below no-context performance, suggesting that retrieval alone is insufficient — fine-tuning on domain-specific noisy retrievals may be necessary.

## 6.2. Without Gold Passages

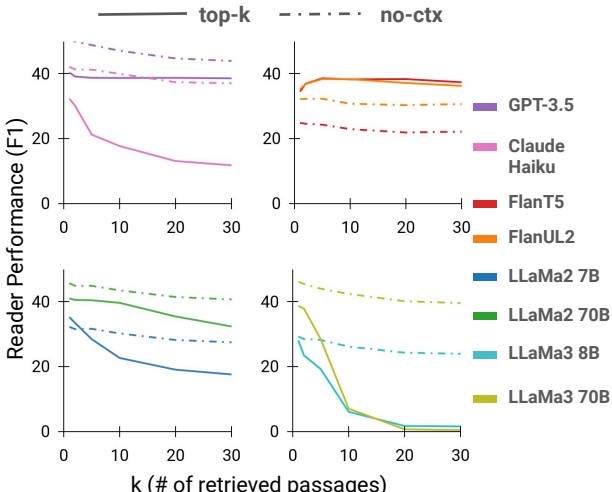

Figure 7: NQ results when no gold passages are retrieved.

When no gold passages are retrieved, most models degrade

below their no-context baseline (Figure 7). This is not surprising since these models are instructed to use the context at hand, which has insufficient information.

What is more interesting is that FLAN models outperform no-context baselines even without gold passages. They seem better than other readers at processing partial clues from these non-gold passages. For example, on NQ with k = 5, the FLAN models achieve 20% accuracy when no gold paragraphs are retrieved but paragraphs from the gold Wikipedia pages are present. Although such paragraphs are not sufficient themselves, they are highly related to the right information, thus providing some contextual clues.

## 6.3. Can Prompting Improve Noise Filtering?

We test whether explicit relevance instructions improve noise filtering (Figure 8). We pick one noise-robust model (FLANT5) and one noise-sensitive model (LLAMA2 7B).

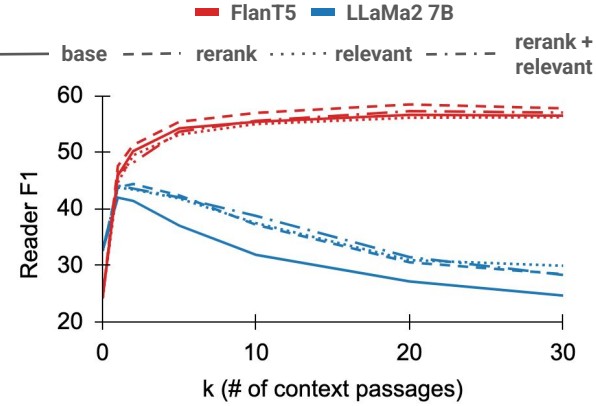

Figure 8: Effects of applying reranking and instructing the model to focus on the relevant passages ("relevant"). These results are for when the retriever is ColBERT and the dataset is NQ. Results for HotpotQA and BioASQ are at Appendix O.

**Prompting Has Mixed Effects and Does Not Improve Stability** Across models, **reranking consistently improves retrieval**, while **prompting has inconsistent effects**. Prompting helps the noise-sensitive model but has no impact on the noise-robust model, which may already have the pretrained ability to pay attention to relevant passages. Interestingly, prompting does not improve performance when reranking is already applied, reinforcing that prompting cannot compensate for poor retrieval, and when retrieval is strong, prompting is redundant.

**Prompting Can Harm Performance in Specialized Domains** In BioASQ, prompting degrades performance likely because the reader is not pretrained on enough domain-specific knowledge to have and context and reason what is relevant or not.

**Key Takeaways** Future RAG research should focus on fine-tuning readers for noise resilience rather than relying on retrieval-side interventions. While prompting can help, it is not a universal fix. RAGGED provides a structured way to assess noise robustness, guiding both retrieval adaptation and reader optimization.

## 7. When Does a Better Retriever Actually Improve RAG Performance?

The reader robustness trends from Section 5 persist across retrievers and even with reranking. That is not to say retriever choice has no impact. Retriever choice still affects retrieval efficiency, computational cost, and domain-specific performance. Below, we compare ColBERT (a neural retriever) and BM25 (a lexical retriever), analyze their impact on different reader types, and assess reranking as a retrieval-side intervention.

### 7.1. When Does a Stronger Retriever Improve RAG Performance?

We evaluate retriever effects using two metrics in Table 8: (1) **Average Difference**: Mean $F_1$ difference between ColBERT and BM25 across $k = 1$ to 50. (2) **Optimal $F_1$ Difference**: The peak performance difference between ColBERT and BM25 at each reader's optimal-$k$. These metrics assess whether a stronger retriever consistently benefits downstream readers (Table 8).

**Retriever Quality Improves Recall, But Not Always Reader Performance** While ColBERT consistently achieves higher recall@k than BM25, its downstream gains vary by reader type. For peak-then-decline models (LLaMa, Claude-3-Haiku), ColBERT performs worse than BM25 at large $k$, despite better retrieval quality. This suggests that some readers are highly sensitive to the nature of the retrieval noise. One possible explanation is that ColBERT retrieves more semantically similar passages, which can be more distracting and misleading than a less relevant passage retrieved from BM25.

**Retriever Improvements Have Modest Gains in Open-Domain QA** Although ColBERT improves recall significantly in NQ (+21.3 recall@k) and HotpotQA (+14.6 recall@k), the corresponding reader gains are much smaller (+5.2 and +1.9 $F_1$, respectively). The low ratio of reader gain to retrieval gain (0.13 in HotpotQA) suggests that better retrieval alone does not guarantee proportionately substantial reader improvement, especially in open-domain settings.

**Specialized Domains Benefit More from Stronger Retrieval** In contrast, in specialized domains (BioASQ), even small retrieval improvements (+0.7 recall@k) yield substantial reader gains (+2.08 $F_1$).

One possible explanation is that domain-specific terminology provides stronger retrieval cues, allowing retrievers to separate relevant from irrelevant content more effectively. This reduces the need for reader-level noise filtering, making retrieval improvements directly beneficial.

### 7.2. Does Reranking Improve Retriever-Reader Alignment?

**Reranking Helps More in Open-Domain QA Than in Specialized Domains** Reranking improves performance in open-domain datasets (NQ, HotpotQA), particularly for noise-sensitive models like LLaMa Appendix O. However, in BioASQ, reranking fails to provide consistent gains and in some cases degrades performance. In open-domain QA, retrieval errors often involve partially relevant passages, meaning reranking can improve performance by elevating the most useful documents. However, in domain-specific tasks like BioASQ, where documents contain dense technical content, reranking may prioritize semantically similar but not specific enough passages, leading to worse performance.

**Reranking Outperforms Prompting, But Gains Do Not Stack** Across datasets, reranking consistently outperforms prompting as a noise-filtering strategy. However, applying both does not yield additional gains. Once retrieval quality is improved via reranking, prompting has little residual effect. Since reranking, in some sense, filters input before it reaches the model, prompting has no additional noise to filter out.

**Key Takeaways** Stronger retrievers do not always lead to better RAG performance, and reader robustness to noise remains the key bottleneck. While dense retrievers improve recall, their benefits depend on how well the reader integrates retrieved information. Reranking, although beneficial, depends on domain-specific retrieval quality and does not fundamentally change a reader's stability and scalability.

## 8. Related Work

**Retrieval Depth and Performance** Prior work offers mixed conclusions on increasing retrieval depth ($k$). Some studies report consistent improvements (Izacard & Grave, 2021), while others find diminishing returns (Liu et al., 2023) or even performance degradation at high $k$ (Cuconasu et al., 2024; Jiang et al., 2024).

Rather than contradictory, we find these trends depend on the reader's robustness to noise. Our work systematically evaluates retrieval depth effects across diverse readers, distinguishing improve-then-plateau vs. peak-then-decline behaviors as key factors in retrieval effectiveness.

**Domain-Specific RAG Effectiveness** RAG's impact varies by domain, particularly for long-tail knowledge. Some studies suggest retrieval is beneficial (Kandpal et al., 2023), while others find it unnecessary or even harmful for common knowledge (Mallen et al., 2023).

We show that domain effects are not inherently stronger or weaker but depend on the retriever-reader interaction, highlighting the need to optimize retrieval strategies per task rather than assuming domain specificity guarantees improvements.

**Retriever Choice and Reader Performance** Dense retrievers often improve retrieval quality (Lewis et al., 2020), but their downstream impact is not always positive. Finardi et al. (2024) report a correlation between retriever and reader performance in specialized settings, yet our results reveal that stronger retrievers do not always yield better RAG outputs, especially for noise-sensitive readers.

While retriever quality enhances recall, reader robustness dictates final performance, with specialized-domain tasks benefiting disproportionately from even minor retrieval improvements. This underscores the need for domain-aware retrieval strategies rather than assuming higher retrieval accuracy guarantees better generation.

## 9. Conclusion

Retrieval-augmented generation (RAG) systems are widely used to enhance language models, but their performance hinges not just on retrieval quality, but on the reader's ability to handle noise and uncertainty. Our study demonstrates that retrieval depth must be dynamically tuned for each model, and that reader robustness, not retriever strength, is the key driver of scalable and stable RAG performance.

To support this insight, we introduce RAGGED, a modular evaluation framework that systematically analyzes retrieval depth, noise sensitivity, and reader-retriever dynamics. Through two new metrics – RAG Stability Score (RSS) and RAG Scalability Coefficient (RSC) – RAGGED offers a principled way to assess how reliably and efficiently models use retrieved information across configurations and domains.

These findings challenge the assumption that retrieval quality alone governs RAG success, and highlight the importance of tuning retrieval strategies around reader behavior. As models continue to evolve, RAGGED remains applicable as a model-agnostic harness for measuring retrieval sensitivity and guiding deployment decisions.

Looking ahead, expanding RAGGED to adversarial, outdated, or temporally shifting noise scenarios will further enhance its relevance to high-stakes, real-world settings. By formalizing how we evaluate reader robustness and retrieval utility, RAGGED lays a foundation for building more reliable, adaptive, and efficient retrieval-augmented generation systems.

## Impact Statement

Our work contributes to improving the evaluation and optimization of RAG systems, which are increasingly used in knowledge-intensive tasks such as question answering, fact-checking, and scientific information retrieval. By introducing a systematic framework for assessing RAG stability and scalability, our study provides actionable insights for building more reliable and robust AI systems.

**Ethical Considerations:** While RAG systems enhance factual accuracy by incorporating external knowledge, they also introduce risks such as information distortion when retrieval is noisy or biased. Our findings highlight the importance of reader robustness to retrieval noise, suggesting that deployments of RAG models should include safeguards against misleading or incorrect retrieved content.

**Future Societal Impact:** As RAG-based models become integral to decision-making in fields like healthcare, law, and education, ensuring their stability and reliability is crucial. The RAGGED framework provides a principled way to measure and improve retrieval robustness via RSS and RSC, which could help mitigate misinformation risks in high-stakes applications.

While our study focuses on evaluation, its insights can inform the development of more scalable and stable RAG systems.

## Acknowledgements

Special thanks to Alex Cabrera, Alex Bäuerle, Jun Araki, Md Rizwan Parvez for providing Zeno support for analysis visualization. Our appreciation extends to Hao Zhu, Jacob Springer, and Vijay Viswanathan for providing feedback for our paper. This paper was supported in part by a gift from Bosch research.

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

# A. Reader Implementation Details

We truncate the context to make sure the the rest of the prompt still fits within a reader's context limit. Specifically, when using FLANT5 and FLANUL2 readers, we use T5Tokenizer to truncate sequences to up to $2k$ tokens; when using LLAMA models, we apply the LlamaTokenizer and truncate sequences by $4k$ tokens for LLAMA2 and $8k$ for LLAMA3. For closed-source models, we spent around $300. Subsequently, we incorporate a concise question-and-answer format that segments the query using "Question:" and cues the model's response with "Answer:", ensuring precise and targeted answers.

For our reader decoding strategy, we used greedy decoding with a beam size of 1 and temperature of 1, selecting the most probable next word at each step without sampling. The output generation was configured to produce responses with 10 tokens. The experiments were conducted on NVIDIA A6000 GPUs, supported by an environment with 60GB RAM. The average response time was ∼1.1s per query when processing with a batch size of 50.

# B. Prompt

For all experiments, we use the following prompt:

> *Instruction:* Give simple short one phrase answers for the questions based on the context
> *Context:* [passage$_1$, passage$_2$, $\cdots$, passage$_k$]
> *Question:* [the question of the current example]
> *Answer:*

For the "relevant" prompt, we swap the instruction for "Give simple short one phrase answers for the questions based on only the parts of the context that are relevant to the question."

# C. Dataset Details

All corpus and datasets use English.

For NQ and HotpotQA datasets in the open domain, we use the Wikipedia paragraphs corpus provided by the KILT benchmark (Petroni et al., 2021). For BioASQ, we use the PubMed Annual Baseline Repository for 2023 (of Medicine, 2023), where each passage is either a title or an abstract of PubMed papers. Dataset sizes are in Table 3.

The Medline Corpus is from of Medicine (2023) provided by the National Library of Medicine.

For NQ and HotpotQA, we use KILT's dev set versions of the datasets, allowed under the MIT License (Petroni et al., 2021). For BioASQ (Krithara et al., 2023), we use Task 11B, distributed under CC BY 2.5 license.

| Corpus | # of par | # of doc | Avg # of doc |
|--------|----------|----------|--------------|
| Wikipedia | 111M | 5M | 18.9 |
| Medline | 58M | 34M | 1.7 |

Table 2: Retrieval corpus information

| Dataset | # of Queries |
|---------|--------------|
| NQ | 2837 |
| HotpotQA | 5600 |
| BioASQ | 3837 |

Table 3: Dataset information

# D. Comparison with No-Context Performance

We include additional reader results comparing ColBERT and BM25 at Table 4 and Table 5.

# E. Relating Reader Trends to Reader Architectures and Training Details

There are two primary types of readers observed in our experiments:

- Peak-then-Decline Behavior: Models including those from the LLAMA and CLAUDE families show sensitivity to noisy documents, leading to performance degradation as the number of retrieved passages (k) increases beyond a certain point.

- Improve-then-Plateau Behavior: Models including those from the GPT and FLAN families are more robust to noise, continuing to benefit from additional context until performance plateaus.

Since we do not have access to the details of the closed-source models, we will focus on providing hypotheses according to the open-source model (LLAMA belonging to the peak-then-decline behavior and the FLAN models belonging to the improve-then-plateau family).

On one hand, FLAN, an improve-then-plateau model family, incorporates additional strategies explicitly designed to handle noisy or diverse contexts. It employs denoising strategies, such as a mixture-of-denoisers, during training to improve its robustness to irrelevant or noisy contexts. These enhancements enable it to filter out noise more effectively.

On the other hand, LLAMA 's training predominantly relies on next-token prediction with limited exposure to noisy or retrieval-specific scenarios, making it sensitive to noise at higher k.

We also note that there are some model architecture features that alone do not determine reader behavior:

| Model | NQ | | HotpotQA | | BioASQ | |
|---|---|---|---|---|---|---|
| | ColBERT | BM25 | ColBERT | BM25 | ColBERT | BM25 |
| GPT-3.5 | 1.1 | -8.0 | 7.7 | 5.1 | 8.9 | 7.4 |
| CLAUDE Haiku | -15.7 | -22.5 | -6.5 | -10.2 | -21.7 | -25.9 |
| FlanT5 | **28.9** | **12.6** | **20.9** | 13.5 | **16.6** | **11.9** |
| FlanUL2 | 21.5 | 4.9 | 18.9 | **15.7** | 5.3 | 2.8 |
| LLaMA 2 7B | 1.4 | -4.5 | 10.4 | 8.2 | 6.4 | 5.9 |
| LLaMA 2 70B | -0.1 | -7.6 | 11.2 | 9.2 | 4.4 | 3.9 |
| LLaMA 3 8B | -13.3 | -14.9 | -6.7 | -5.7 | -12.1 | -14.9 |
| LLaMA 3 70B | -24.9 | -26.2 | -12.0 | -11.3 | -18.0 | -21.4 |
| **Average (per dataset)** | -0.1 | -8.3 | 5.5 | 3.06 | -1.3 | -3.8 |

Table 4: The average difference between the $F_1$ score of RAG with $k$ passages from ColBERT or BM25 and the $F_1$ score of no-context generation, calculated across $k$ values from 1 to 50 for each dataset. Each value represents the difference between the $F_1$ score of the reader+retriever combination and the $F_1$ score of the reader alone (without RAG or context).

| Model | NQ | | HotpotQA | | BioASQ | |
|---|---|---|---|---|---|---|
| | ColBERT | BM25 | ColBERT | BM25 | ColBERT | BM25 |
| GPT-3.5 | 3.8 | -3.1 | 8.8 | 7.3 | 10.9 | 10.6 |
| CLAUDE Haiku | -2.4 | -14.9 | 3.9 | 0.7 | -1.7 | -8.2 |
| FlanT5 | **32.5** | **22.6** | **23.5** | **20.4** | **18.0** | **13.0** |
| FlanUL2 | 24.4 | 14.8 | 22.0 | 19.7 | 7.1 | 3.4 |
| LLaMA 2 7B | 9.7 | -0.5 | 15.2 | 11.0 | 8.5 | 6.9 |
| LLaMA 2 70B | 4.3 | -0.3 | 14.0 | 11.4 | 7.3 | 6.8 |
| LLaMA 3 8B | 3.9 | -3.0 | 11.2 | 8.4 | 3.6 | 1.7 |
| LLaMA 3 70B | -0.7 | -9.6 | 14.9 | 8.1 | 4.4 | 0.3 |
| **Average (per dataset)** | 9.44 | 0.8 | 14.2 | 10.9 | 7.3 | 4.3 |

Table 5: The difference between the $F_1$ score of RAG optimal $k^*$ from ColBERT or BM25 and the $F_1$ score of no-context generation. Each value represents the difference between the $F_1$ score of the reader+retriever combination at optimal $k^*$ and the $F_1$ score of the reader alone (without RAG or context).

- Context window size: Models with longer context limits like LLaMA 2 (4k tokens) don't necessarily process a larger number of contexts better than models with smaller context limits like FLAN (2k tokens).

- Encoder-decoder v. decoder: LLaMA is a decoder-only model that displays peak-then-decline behavior, but GPT models are also decoder-only and instead display improve-then plateau behavior.

## F. Preliminary Results on CRAG

To evaluate the generalization of our findings to more recent RAG benchmarks, we conducted a preliminary study using the CRAG dataset (Yang et al., 2024). We selected one representative model from each of the two major reader behavior classes: FLANT5 (improve-then-plateau) and LLAMA2 (peak-then-decline).

As shown in Table 6, the core reader trends persist: LLAMA2 exhibits early performance saturation and plateaus, while FLANT5 demonstrates a mild performance peak followed by flattening. Although performance is lower overall (as expected given CRAG's difficulty), these results suggest that RAGGED's retrieval-depth insights can generalize to this newer benchmark.

Table 6: $F_1$ scores of FLANT5 and LLAMA2 on CRAG at varying retrieval depths (k).

| Model | $k = 1$ | $k = 5$ | $k = 10$ | $k = 20$ | $k = 30$ |
|---|---|---|---|---|---|
| FLANT5 | 0.19 | 0.17 | 0.18 | 0.18 | 0.18 |
| LLAMA2 | 0.20 | 0.23 | 0.23 | 0.23 | 0.23 |

## G. Investigating FLANT5 's High RSS and Input Truncation

In Figure 5, FLANT5 achieves an RSS of 0.99 on the NQ dataset. One possibility is that truncation effects at higher retrieval depths (e.g., $k = 25$) may mask additional context, artificially inflating stability.

To assess this, we compared tokenized input lengths be-

tween $k = 20$ and $k = 25$. In 32% of cases, the $k = 25$ input included more tokens than $k = 20$, indicating that additional retrieved passages were indeed processed. Despite this, the model's $F_1$ score changes by $< 0.5$ on average, supporting our interpretation that FLANT5's high RSS reflects genuine retrieval-depth robustness, not an artifact of context truncation.

## H. Slice Analysis on Other Datasets

We include *with*-gold-passages results for HotpotQA at Figure 9 and for BioASQ at Figure 10.

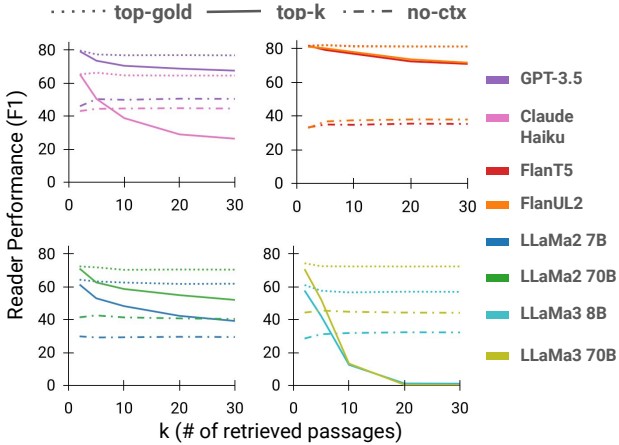

Figure 9: HotpotQA results when there is sufficient information (all gold passages) included in the top-k passages to answer the question. For multi-hop questions, we select examples retrieved with all gold passages within the top-$k$ passages since all passages are necessary to answer the question.

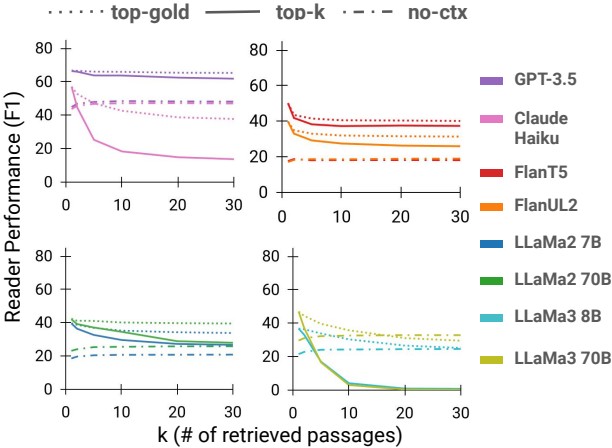

Figure 10: BioASQ results when there is sufficient information (at least one gold passage) included in the top-k passages to answer the question.

We include *without*-gold-passages results for HotpotQA at Figure 11 and for BioASQ at Figure 12.

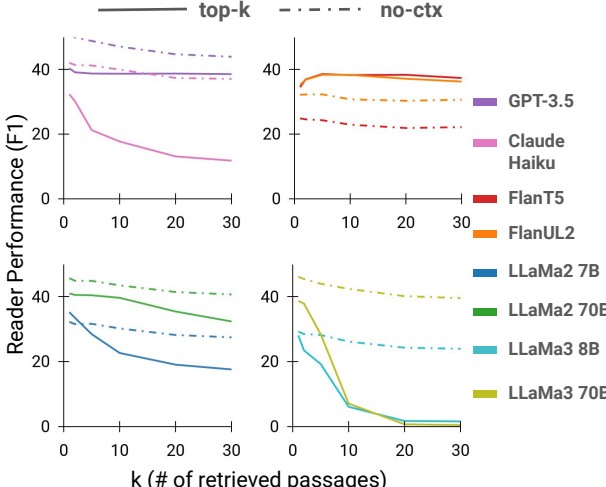

Figure 11: HotpotQA results when there are no gold passages included in the top-k passages to answer the question.

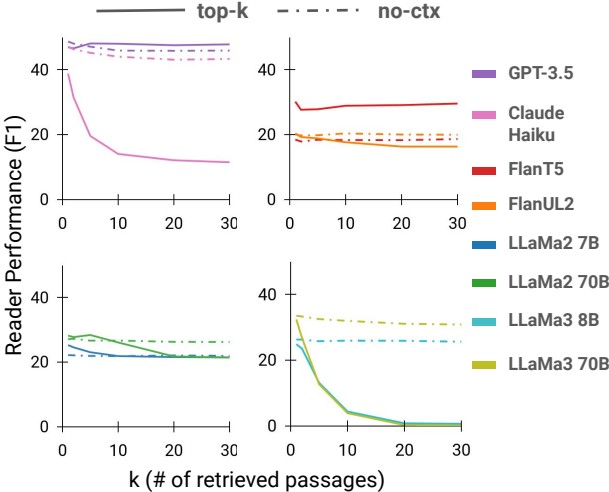

Figure 12: BioASQ results when there are no gold passages included in the top-k passages to answer the question.

## I. Comparing Optimal k Values

We include the optimal $k$ for ColBERT and BM25 in Table 7.

## J. LLM-Based Evaluation

While we chose $F_1$ for its simplicity and alignment with prior work, we agree that it may not fully reflect nuanced semantic equivalence. To address this, we ran an LLM-based evaluation of the models for the NQ dataset using

| Model | NQ | | HotpotQA | | BioASQ | | Average (per reader) | |
|---|---|---|---|---|---|---|---|---|
| | BM25 | ColBERT | BM25 | ColBERT | BM25 | ColBERT | BM25 | ColBERT |
| GPT-3.5 | 50 | 20 | 50 | 20 | 20 | 20 | 40 | 20 |
| CLAUDE Haiku | 1 | 1 | 1 | 1 | 1 | 1 | 1 | 1 |
| FlanT5 | 50 | 20 | 10 | 10 | 50 | 1 | 36.67 | 10.33 |
| FlanUL2 | 50 | 10 | 20 | 10 | 2 | 1 | 24 | 7 |
| LLAMA 2 7B | 1 | 1 | 2 | 2 | 2 | 1 | 1.67 | 1.33 |
| LLAMA 2 70B | 10 | 5 | 10 | 2 | 5 | 5 | 8.33 | 4 |
| LLAMA 3 8B | 1 | 1 | 1 | 1 | 1 | 1 | 1 | 1 |
| LLAMA 3 70B | 1 | 1 | 1 | 1 | 1 | 1 | 1 | 1 |
| Average (per dataset) | 20.5 | 7.38 | 11.88 | 5.88 | 10.25 | 3.88 | 14.21 | 5.71 |

Table 7: Optimal $k^*$ for BM25 and ColBERT (NQ, HotpotQA, and BioASQ).

Prometheus (Kim et al., 2024), specifically the Prometheus-7b-v2.0 model. We find that the conclusions about reader trends do not change: the same reader trends apply to the same models (peak-then-decline v. improve-then-plateau). We use Prometheus-7b-v2.0 to evaluate the correctness of the generated answer against the gold answer on a 5-point scale, where 1 is the least correct and 5 is the most correct Figure 13.

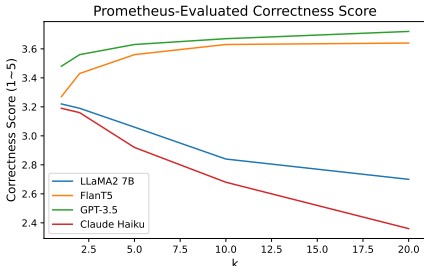

Figure 13: Reader Performance on NQ dataset as evaluated by Prometheus on a 5-point scale where 1 is the least correct and 5 the the most correct.

## K. Comparing Reader Trends when using ColBERT v. BM25

We include the top-k performance for ColBERT, BM25 Figure 14 and in Table 8

## L. Comparing Neural Retrievers

We compare the top-$k$ performance of ColBERT and Contriever at Figure 15.

## M. Comparing GPT-3.5 and GPT-4O

We compare how GPT-3.5 and GPT-4O perform, and find that they both display the same reader trend of improve-then-plateau, with the main difference being GPT-4O's reader performance is shifted up (Figure 16).

## N. Retriever Performance

We include the retriever performance at select $k$'s at Table 9.

## O. Effect of Reranker and Relevance Prompting

We test whether reranking and/or explicit relevance prompting instructions improve noise filtering on NQ, HotpotQA, and BioASQ (Figure 8, Figure 17, Figure 18). We pick one noise-robust model (FLANT5) and one noise-sensitive model (LLAMA2 7B) to demonstrate preliminary results.

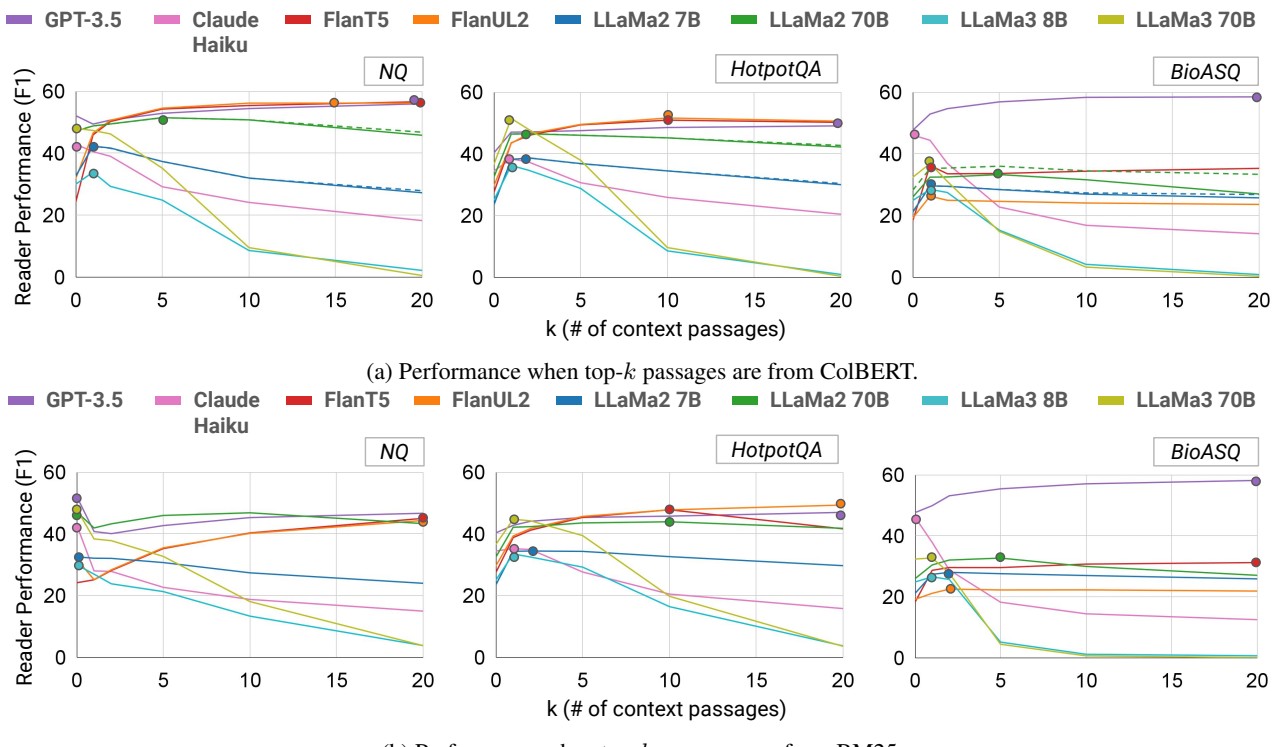

(a) Performance when top-$k$ passages are from ColBERT.

(b) Performance when top-$k$ passages are from BM25.

Figure 14: Top-k performance on NQ, HotpotQA, and BioASQ. Colored circles mark the reader performance at optimal $k^*$.

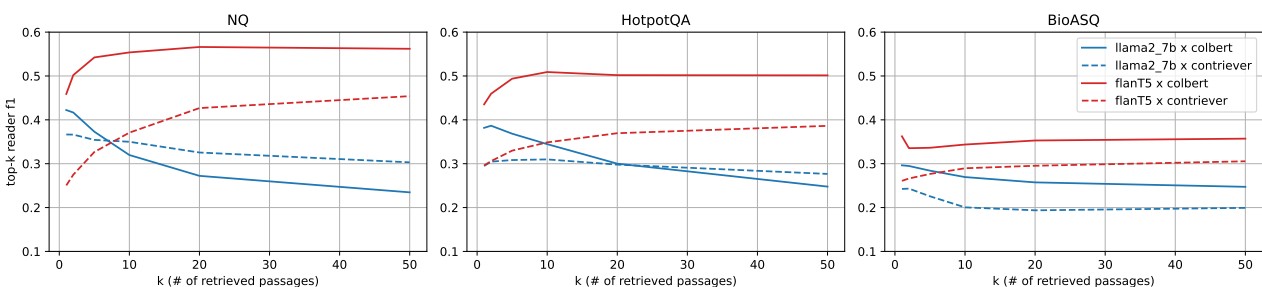

Figure 15: Example of how reader response to increasing context applies across neural retrievers (e.g., ColBERT and Contriever) and datasets. We choose one reader model from each trend for demonstration — LLAMA2 7B for peak-then-decline and FLANT5 for improve-then-plateau.

| Model | Average Difference (across $k$) | | | Difference in Optimal Performance | | |
|---|---|---|---|---|---|---|
| | NQ | HotpotQA | BioASQ | NQ | HotpotQA | BioASQ |
| GPT-3.5 | 8.6 | 2.0 | 1.1 | 6 | 1 | 0 |
| Claude Haiku | 3.9 | 4.0 | 2.4 | **12** | 3 | **6** |
| FlanT5 | 12.6 | **10.5** | **4.2** | 9 | 3 | 4 |
| FlanUL2 | **12.9** | 2.0 | 1.9 | 9 | 2 | 3 |
| LLaMa2 7B | 3.6 | 0.9 | -0.3 | 10 | 4 | 1 |
| LLaMa2 70B | 2.6 | 0.7 | -0.2 | 4 | 2 | 0 |
| LLaMa3 8B | -0.7 | -2.2 | 1.4 | 6 | 2 | 1 |
| LLaMa3 70B | -1.9 | -2.7 | 1.5 | 8 | **6** | 4 |
| **Average** | 5.2 | 1.9 | 1.5 | 8 | 2.9 | 2.4 |

Table 8: For each reader, the average difference and optimal difference in $F_1$ scores between ColBERT and BM25 are reported. (See the main text above for detailed definitions.)

| Retriever | Recall@k | | | | | |
|---|---|---|---|---|---|---|
| | 1 | 2 | 5 | 10 | 20 | 50 |
| *NQ* | | | | | | |
| BM25 | 2.7 | 4.4 | 8.0 | 11.5 | 16.3 | 22.8 |
| | 10.3 | 16.3 | 27.8 | 36.8 | 47.7 | 53.2 |
| ColBERT | 12.3 | 18.0 | 25.7 | 32.1 | 38.1 | 41.8 |
| | 27.2 | 38.8 | 54.4 | 65.0 | 72.9 | 77.2 |
| Contriever | 4.65 | 6.91 | 11.14 | 15.17 | 20.19 | 28.46 |
| | 24.0 | 32.3 | 44.9 | 53.2 | 62.1 | 72.0 |
| *HotpotQA* | | | | | | |
| BM25 | 19.1 | 25.9 | 34.6 | 41.1 | 46.8 | 54.2 |
| | 23.3 | 31.2 | 42.7 | 52.1 | 59.1 | 62.8 |
| ColBERT | 31.1 | 40.1 | 49.9 | 56.2 | 61.9 | 64.9 |
| | 34.2 | 44.7 | 56.3 | 63.6 | 69.9 | 73.1 |
| Contriever | 2.35 | 4.44 | 8.14 | 11.75 | 15.46 | 20.79 |
| | 22.39 | 29.54 | 39.39 | 45.71 | 51.51 | 59.08 |
| *BioASQ* | | | | | | |
| BM25 | 8.8 | 12.9 | 19.6 | 25.8 | 33.3 | 37.8 |
| | 12.4 | 16.4 | 23.9 | 30.6 | 38.7 | 43.6 |
| ColBERT | 8.8 | 13.5 | 20.7 | 27.1 | 34.3 | 38.6 |
| | 14.2 | 18.2 | 25.6 | 32.2 | 39.8 | 44.2 |
| Contriever | 3.82 | 5.87 | 9.55 | 12.95 | 17.48 | 24.58 |
| | 7.91 | 10.55 | 15.36 | 19.61 | 24.89 | 33.03 |

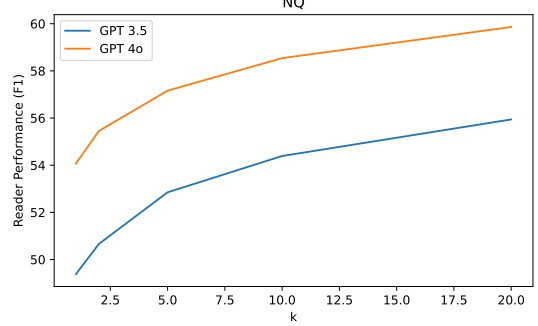

Figure 16: Comparison of GPT-3.5 and GPT-4O performance on NQ.

Table 9: Retriever performance (recall@k). For the Wikipedia-based dataset, the top row indicates recall@k at the retrieval unit of Wikipedia paragraph and the bottom row for the unit of Wikipedia page. For BioASQ, the top row indicates recall@k at the unit of title or abstract of a PubMed article and the bottom row at the unit of the article itself.

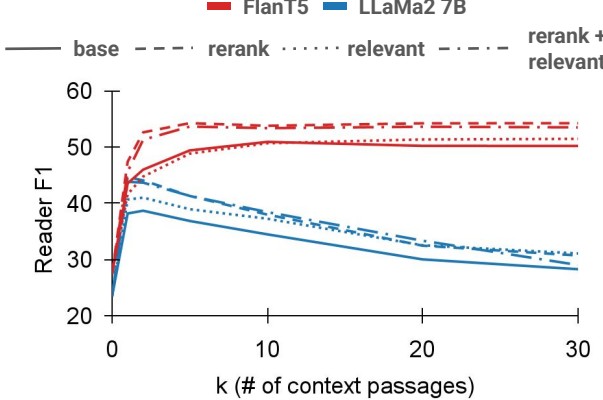

Figure 17: Effects of applying reranking and instructing the model to focus on the relevant passages ("relevant"). These results are for when the retriever is ColBERT and the dataset is HotpotQA.

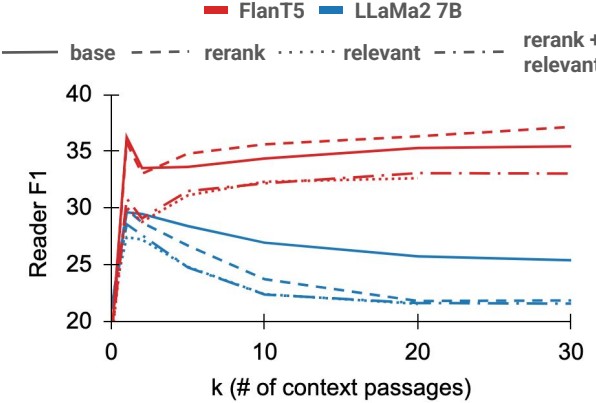

Figure 18: Effects of applying reranking and instructing the model to focus on the relevant passages ("relevant"). These results are for when the retriever is ColBERT and the dataset is BioASQ.

