# OpenReview forum: "RAGGED: Towards Informed Design of Scalable and Stable RAG Systems"
_ICML.cc/2025/Conference — ICML 2025 poster_

### Official Review · Reviewer_SdGE · 2025-03-11

**Overall Recommendation:** 2

**Summary:**

This paper carries out empirical analysis to shed light on the impact of retrieval in a RAG system: 1/ when retrieval is needed, 2/ impact of retrieval depth, 3/ noisy retrieval, 4/ relation between retrieval improvements to final performance improvements. The paper proposes two metrics, RAG Stability Score and RAG Scalability Coefficient, to measure the robustness of a RAG system. Experiments are carried out on NQ, HotpotQA, and BioASQ dataset. The main findings show that: 1/ retrieval may not help depending on if model is robust to noise, 2/ when increasing retrieval depth some models improve-then-plateau some models peak-then-decline, 3/ model noise robustness is more important than retrieval noise filtering, 4/ retriever improvements does not always lead to better response quality.

**Claims And Evidence:**

This is an empirical paper where the findings are model-dependent. The claims made in the submission are supported by clear and convincing evidence.

**Essential References Not Discussed:**

No

**Experimental Designs Or Analyses:**

Dataset: Experiments are carried out on NQ, HotpotQA, and BioASQ datasets. However, I think NQ and HotpotQA are not diverse or representative enough for RAG evaluation due to many facts are memorized by LLMs.

Evaluation metric: unigram-based score may not reliably measure LLM performance due to challenges in lexical matching.

**Methods And Evaluation Criteria:**

I have the following concerns over the proposed metrics:

1. The proposed RAG Scalability Score (RSS) is defined as a symmetric measure, yet retrieval results are inherently asymmetric. Removing higher-ranked retrieval results has a greater negative impact than adding lower-ranked ones. Given this asymmetry, it is unclear why RSS can be considered a reliable and stable metric.

2. I find the RAG Scalability Coefficient (RSC) is difficult to interpret. It is defined as the product of the last retrieval depth before performance plateaus and the cumulative improvements, but the rationale behind this multiplicative relationship is unclear. Additionally, this metric depends on a hyperparameter $\epsilon$, which may be challenging to tune in practice.

**Other Comments Or Suggestions:**

Minor suggestion: it is recommended to produce figures in vector format, e.g., pdf.

**Other Strengths And Weaknesses:**

I liked the empirical findings and insights from this paper. But I think the proposed metrics are not well-defined and the datasets may not cover representative RAG use case.

**Questions For Authors:**

Your findings appear to be closely tied to the specific dataset or model used. How do you envision these insights applying to newer LLMs to time-sensitive questions, and what factors might influence their generalizability?

**Relation To Broader Scientific Literature:**

This paper offers empirical findings and suggestions on developing RAG systems. The evaluation setups are similar to literature and the paper proposes two new metrics, RAG Stability Score and RAG Scalability Coefficient, to measure the robustness of a RAG system.

**Theoretical Claims:**

Not Applicable

---

> ### Author Rebuttal · Authors · 2025-04-01
>
> We thank the reviewer for the thoughtful and constructive feedback. We appreciate the recognition of our empirical contributions and insights into retrieval dynamics, particularly the introduction of RSS and RSC as tools for understanding reader robustness and retrieval scalability.
>
> **1. Symmetry of RSS**
>
> Thank you for the insightful observation. While it is true that adding lower-ranked documents (right side) and removing top-ranked ones (left side) may have different theoretical impacts, our empirical results show that this asymmetry is not consistent in practice. For example, LLAMA2-7B degrades more on the left (removing documents), whereas LLAMA3-70B degrades more on the right (adding documents). This variation supports our use of a symmetric window, which offers a model-agnostic and balanced way to measure stability. We will clarify this in the revision.behaviors. We will clarify this rationale in the revised manuscript.
>
>
> **2. Clarifying Design of RSC**
>
> RSC is defined as the product of two complementary components: 1) the last retrieval depth before plateau/decline, which captures how far the model scales before benefits taper off, 2) the cumulative performance gain up to that point, which quantifies the total utility derived from increasing retrieval.
>
> By multiplying these factors, RSC distinguishes clearly between models that achieve quick but limited improvements (early plateau) and those that sustain meaningful improvements over deeper retrieval depths.
>
> The hyperparameter ε, it is intentionally user-defined, similar to early stopping in training. It allows users to specify what constitutes a meaningful gain, based on their deployment trade-offs. We chose ε = 0.5 for RSC based on empirical analysis: the standard deviation of F1 across all models and k values is consistently below 0.5 (mean ≈ 0.38, max = 0.46). This makes 0.5 a conservative but meaningful threshold that exceeds metric noise. Moreover, increasing ε to 0.6 or 0.7 preserves model ranking, indicating robustness to the threshold choice.
>
> **3. F1 vs. Semantic Metrics**
>
> We recognize that unigram-based F1 may underrepresent semantic correctness. To address this, we validated our findings using an LLM-based semantic correctness metric on a subset of responses (Appendix H), and found strong alignment with the trends captured by F1. We will make this point clearer in the revised version.
>
> **4. Memorization of Dataset by LLMs.**
>
> Thank you for the comment. We agree that newer datasets reduce the risk of memorization. Our original evaluation includes NQ (2019), HotpotQA (2018), and BioASQ (2023) to cover open-domain, multi-hop, and biomedical QA settings.
> To extend this, we incorporate a 2024 dataset, CRAG, as suggested by reviewer jqB4. We ran preliminary experiments on CRAG using two representative readers: FLAN-T5 (of improve-then-plateau behavior) and LLAMA-2 (of showing peak-then-decline). These behavioral trends remain consistent on CRAG, suggesting that the retrieval-depth dynamics generalize well to newer datasets. We will include full CRAG results in the camera-ready version.
>
> | Model     | k=1   | k=5   | k=10  | k=20  | k=30  | k=50  |
> |-----------|-------|-------|-------|-------|-------|-------|
> | FLAN-T5   | 0.190 | 0.174 | 0.178 | 0.175 | 0.175 | 0.175 |
> | LLAMA     | 0.204 | 0.227 | 0.227 | 0.227 | 0.227 | 0.227 |
>
> **4. Applicability to Newer LLMs and Time-Sensitive Queries**
>
> We conducted additional experiments on dynamic questions in CRAG using both FLAN-T5 and LLAMA, and observed the following:
>
> - FLAN-T5 maintains its improve-then-plateau retrieval-depth behavior on dynamic questions, consistent with its pattern on static ones. This suggests strong generalization and robustness to retrieval noise across domains.
>
> - LLAMA, in contrast, shifts from a peak-then-decline trend on static questions to an improve-then-plateau trend on dynamic questions. We attribute this to differences in internal knowledge: on static questions, LLAMA already knows the answer and retrieval introduces redundancy or contradictions, degrading performance at higher depths. On dynamic questions—where LLAMA lacks internal knowledge—it relies more heavily on retrieved context, and even noisy documents provide helpful signal, leading to continued improvement.
>
> This contrast reinforces the value of our reusable framework: even when retrieval dynamics vary across models or tasks, RSS and RSC offer a principled way to detect, quantify, and interpret these differences. Rather than assuming fixed behavior, our reusable framework and metrics help uncover how model robustness and scalability shift across domains.
>
> Wwe expect newer LLMs—particularly those fine-tuned for retrieval-augmented tasks—to exhibit higher RSS and RSC due to improved handling of noisy or diverse context. Regardless of the model architecture, however, we believe our metrics remain essential tools for diagnosing retrieval sensitivity and informing robust, real-world RAG deployment.

---

> > ### Comment · Reviewer_SdGE · 2025-04-06
> >
> > I'd like to thank the authors for their detailed response, which has addressed some of my concerns. I liked the paper’s insightful empirical findings; however, I believe that the proposed metrics (RSS and RSC) should be refined to better capture retrieval asymmetry and robustness to variances of the evaluation results. I also appreciate the new results for Llama2, but given the rapidly evolving landscape of LLMs, it is uncertain how these empirical findings presented in the paper remain relevant over time.

---

> > > ### Author Response · Authors · 2025-04-07
> > >
> > > We sincerely thank the reviewer for their thoughtful follow-up and for acknowledging the insightful findings of our work.
> > >
> > > **1. Symmetry of RSS**
> > >
> > > Thank you for raising this important point. While RSS currently uses a symmetric window to measure stability around the optimal retrieval depth, we agree that directional sensitivity can be important in practice. For example, some applications may face higher risk from over-retrieval (e.g., latency, cost, or noise from irrelevant documents), while others may be more vulnerable to under-retrieval due to conservative defaults.
> > >
> > > A directional variant of RSS (e.g., reporting LHS-RSS and RHS-RSS separately) could help identify whether a model is more fragile to adding versus removing context. This would allow practitioners to tune systems more cautiously based on which side is riskier, or to design guardrails accordingly.
> > >
> > > That said, our empirical findings show that this asymmetry is not consistent across models, which motivates the use of a symmetric metric as a general-purpose diagnostic. Still, we agree that directional extensions are a valuable refinement and will note this as future work in the revised paper.
> > >
> > > **2. Robustness to Evaluation Variance**
> > >
> > > To assess the sensitivity of our metrics, we computed the standard deviation of F1 across all models and k values (mean ≈ 0.38, max ≈ 0.46). We also verified that varying the key hyperparameters: ε for RSC (e.g., from 0.5 to 0.7) and δ for RSS (e.g., ±5 vs ±10) does not affect model rankings. This supports the robustness of our findings and shows that the metrics are not overly sensitive to reasonable shifts in the evaluation setup. We will make this variance analysis explicit in the final version.
> > >
> > > **3. Generalization to Future LLMs**
> > >
> > > We appreciate the reviewer’s concern about evolving model capabilities. Our primary contribution is a reusable, model-agnostic evaluation framework—not a fixed set of empirical conclusions. RAGGED is designed to support the ongoing evaluation of emerging models by quantifying stability, scalability, and absolute performance across retrieval depths, question types, and domains. As systems change, these new metrics remain valuable tools for uncovering new trends, brittleness, or failure modes.
> > >
> > > At the same time, RAGGED has already surfaced practical, generalizable insights. Across diverse datasets and model families, we consistently find that reader robustness, not retriever strength, is the dominant factor driving RAG trends in scalability and stability. This trend holds even on the recent CRAG (2024) benchmark. As RAG systems continue to evolve, we believe tools like RAGGED are essential for evaluating new models, identifying where robustness breaks down, and guiding improvements in system design.

---

### Official Review · Reviewer_jqB4 · 2025-03-14

**Overall Recommendation:** 3

**Summary:**

The paper presents RAGGED—a evaluation harness for retrieval-augmented generation (RAG) systems. The authors identify conflicting narratives in the previously published literature, and aim to resolve it especially around sensitivity to irrelevant documents. They examine how different retriever methods (e.g., BM25, ColBERT, Contriever) and reader models interact across various tasks and datasets (NQ, HotpotQA, and BioASQ). The framework introduces novel metrics—the RAG Stability Score (RSS) and the RAG Scalability Coefficient (RSC)—to quantify how stable and scalable a model’s performance is as retrieval depth increases. Ultimately, the paper argues that the robustness of the reader to noisy retrieval (i.e., its ability to filter out irrelevant passages) is the key determinant of overall RAG effectiveness.

**Claims And Evidence:**

Reader Robustness Is Paramount: The paper claims that a reader’s ability to handle noise is more critical than mere retriever quality for achieving stable and scalable RAG performance.

Retrieval Depth Effects Are Model-Dependent: It argues that while some models benefit from more retrieved passages (improve-then-plateau behavior), others degrade when faced with increased noise (peak-then-decline behavior).

Retrieval Improvements Yield Nonlinear Gains: Enhanced retriever performance (for instance, using dense retrievers) does not always translate into proportional improvements in reader performance, particularly for noise-sensitive readers.

Domain-Specific Nuances Matter: The dynamics of retrieval and reading differ between open-domain tasks (like NQ and HotpotQA) and specialized domains (such as BioASQ), highlighting the need for tailored configurations.

I find all the claims to be well supported with breadth but not with depth. For example, BM-25, contriever are both not SOTA for embeddings.
1. I would urge the authors to also evaluate their system on modern embeddings such as text-embedding-large (fromOAI) for closed source among many others, and some of the SOTA generative or discriminative models from HuggingFace mteb leaderboard (for opensource).
2. The datasets used by the authors are not the ideal use-cases to evaluate LLM's ability to retrieve. For example, HotPotQA documents are all within a few hundred to thousand tokens with very clear "knowledge" component which most models listed have seen before (given the high overlap with refinedweb and other pre-training datasets). To the extend HotPot pass@K is today used as a pre-training metric. Hence, I would encourage the authors to engage with newer RAG benchmarks such as CRAG

**Essential References Not Discussed:**

Reference of CRAG for evals: https://arxiv.org/html/2406.04744v1

**Experimental Designs Or Analyses:**

I have already mentioned my concerns around the dataset used for eval, and the retrievers.

I would also like to add that the LLMs used are quite dated. However, while I encourage the authors to consider updating this for camera-ready or a subsequent submission, I do NOT think this is a serious concern since it should not be on the authors to keep up with the fast moving model releases and the trends hold independent of the models.

**Methods And Evaluation Criteria:**

Addressed above extensively

**Other Comments Or Suggestions:**

N/A

**Other Strengths And Weaknesses:**

Addressed above

**Questions For Authors:**

In figure 5, flan-T5 RSS is at 0.99. Do you suspect this is a function of the eval setting (context length cap), or do the authors happen to have any additional insights?

**Relation To Broader Scientific Literature:**

In my view this paper answers some of the open questions from the "Lost in the middle " paper by clearly defining the 2 new metrics: RSS and RSC.

**Theoretical Claims:**

No

---

> ### Author Rebuttal · Authors · 2025-04-01
>
> We sincerely thank the reviewer for the constructive and thoughtful feedback. We appreciate that the reviewer recognizes that the paper resolves open questions in existing works and finds all the claims to be well supported.
>
> **1. Expanding to Newer Embedding Models**
>
> We appreciate the suggestion to evaluate newer embedding models. We are in the process of integrating models from the MTEB leaderboard, including leading open-source options like Linq-Embed-Mistral. However, due to the scale of the corpus (~111M passages) and the limited resources and time, we were unable to complete these runs within the rebuttal period. We are actively running the experiments and will include updated results in the revision. We agree that this extension will further strengthen our analysis.
>
>
> **2. Evaluating with CRAG**
>
> Thank you for highlighting the importance of more modern evaluation datasets. We agree that CRAG offers a more recent and challenging benchmark for RAG evaluation.
>
> To assess generalization, we ran preliminary experiments on CRAG during the rebuttal window using two representative readers: FLAN-T5 (of improve-then-plateau behavior) and LLAMA-2 (of showing peak-then-decline). These behavioral trends remain consistent on CRAG, suggesting that the retrieval-depth dynamics generalize well to newer datasets. One difference is that the overall performance is worse (shifted down) compared to the performance we observed in our paper, which is as expected since this is a newer dataset. We will include full CRAG results in the camera-ready version.
>
> | Model     | k=1   | k=5   | k=10  | k=20  | k=30  | k=50  |
> |-----------|-------|-------|-------|-------|-------|-------|
> | FLAN-T5   | 0.190 | 0.174 | 0.178 | 0.175 | 0.175 | 0.175 |
> | LLAMA     | 0.204 | 0.227 | 0.227 | 0.227 | 0.227 | 0.227 |
>
>
> **3. In Figure 5, FLAN-T5 RSS is at 0.99. Do you suspect this is a function of the eval setting (context length cap), or do the authors happen to have any additional insights?**
>
>  Thank you for this insightful question. We investigated whether FLAN-T5’s high RSS score might be influenced by context length limitations, particularly around the optimal retrieval depth of k = 25.
>
> To assess this, we compared token lengths after truncation between k = 20 and k = 25. We found that in at least 32% of examples, the input at k = 25 includes more tokens post-truncation than at k = 20, confirming that additional retrieved content is being processed. Despite this added context, the performance difference between k = 20 and k = 25 is < 0.5 points, suggesting the model remains robust even as more content is introduced.
>
> This supports our interpretation that FLAN-T5’s high RSS is not an artifact of truncation shielding the model from additional noise, but rather reflects genuine retrieval-depth stability. We will include this clarification and supporting analysis in the revised manuscript.

---

### Official Review · Reviewer_eNcB · 2025-03-14

**Overall Recommendation:** 2

**Summary:**

The paper introduces RAGGED, a framework for evaluating Retrieval-Augmented Generation (RAG) systems. It emphasizes that RAG's performance depends not only on retrieval quality but also on the reader's robustness to noise. The study shows that reader robustness is the key factor for RAG stability and scalability, and that fine-tuning readers is more important than improving retrievers. RAGGED provides a structured way to optimize retrieval strategies and guide future research on developing robust, scalable RAG systems.

**Claims And Evidence:**

The RAG Stability Score (RSS) measures stability across retrieval depths, but stability is a complex metric. The paper assumes stability directly correlates with system performance, yet it doesn't address whether improved stability at a specific depth always leads to better task performance or just more predictable behavior. How does RSS capture true model performance beyond retrieval consistency? Can models with lower RSS but higher task-specific performance still be considered stable or robust?

**Essential References Not Discussed:**

No

**Experimental Designs Or Analyses:**

The experiments show variability in performance based on the reader-retriever combination (e.g., GPT-3.5 vs. CLAUDE vs. FLAN). While the inclusion of a range of models is a strength, it’s unclear how the variability between models is accounted for in the analysis. For example, a model like FLAN may benefit from retrieval more consistently than GPT-3.5, but are these differences purely due to the models’ architectures, or are there other factors like training data or task-specific tuning influencing the outcomes?

**Methods And Evaluation Criteria:**

The RAG Stability Score (RSS) and RAG Scalability Coefficient (RSC) are introduced to measure stability and scalability across different retrieval depths. While these metrics provide some insight, they focus heavily on retrieval depth rather than directly measuring task-specific performance. Are these metrics comprehensive enough to evaluate the overall effectiveness of RAG systems in real-world tasks, especially when task-specific performance is more critical than retrieval stability?

**Other Comments Or Suggestions:**

Figures 1 and 8 are not clear enough. Letters in Figure 1 are too small.
The use of ** for bolding in the first paragraph of Section 2 of the article is not standardized.

**Other Strengths And Weaknesses:**

This paper introduces the novel RAGGED framework for evaluating Retrieval-Augmented Generation (RAG) systems, focusing on reader robustness and retrieval depth. The authors provide a significant contribution by developing the RAG Stability Score (RSS) and RAG Scalability Coefficient (RSC), offering a structured way to assess model performance across different configurations. Their extensive empirical analysis across multiple datasets and models strengthens the paper’s validity and relevance to future RAG research.

However, the paper lacks a theoretical foundation to explain its empirical findings, limiting the generalization of its results. Additionally, the discussion of key related works, such as REALM and FiD, is insufficient, and the evaluation metrics focus more on stability than end-task performance.

**Questions For Authors:**

Please refer to the above comments.

**Relation To Broader Scientific Literature:**

The key contributions of the paper extend the findings of prior work by highlighting the nuanced relationship between retrieval depth, model stability, and task-specific effectiveness. It provides a more refined view of RAG, emphasizing the importance of reader robustness and domain-aware retrieval strategies, while challenging the oversimplified notion that stronger retrieval always leads to better performance. These contributions push the research further in understanding and optimizing the interaction between retrievers and readers for RAG systems.

**Theoretical Claims:**

The authors discuss the effects of retrieval depth on RAG performance, but without theoretical analysis, it remains unclear why these effects hold or how to generalize the findings beyond the empirical settings. The lack of theoretical grounding makes it harder to predict the behavior of RAG systems in novel conditions or tasks not covered in the paper.

---

> ### Author Rebuttal · Authors · 2025-04-01
>
> We thank the reviewer for the thoughtful and constructive feedback. We appreciate that the reviewer recognizes this paper provides a significant contribution by introducing structured metrics and is backed by extensive empirical analysis across multiple datasets and models.
>
> **1. RSS and its Relationship to Task-Specific Performance**
>
> Indeed, RSS is not intended to capture absolute task performance, and we were careful not to make such a claim. Instead, it measures the stability of performance near the optimal retrieval depth—capturing sensitivity to retrieval parameter changes, not performance magnitude. As the reviewer notes, these are distinct dimensions of system behavior.
>
> **2. Metric Comprehensiveness and Practical Relevance**
>
> While absolute performance is essential, stability and scalability are equally critical in real-world deployment:
>
> - Stability (RSS) robustness to retrieval depth variation, which is particularly important in real-world scenarios where precise tuning is costly.
>
> - Scalability (RSC) reflects how well a model benefits from deeper retrieval, which is especially important when relevant information is sparse or buried.
>
> Thus, while task-specific performance is foundational, it does not provide a complete picture of real-world robustness or efficiency. We believe that stability, scalability, and absolute performance together form a more holistic and practical evaluation of RAG systems. We also discuss the importance of absolute task performance in sections 4, 6, 7, paying attention to compare with-context and no-context performance to see when RAG actually helps.
>
> **3. Can models with lower RSS but higher task-specific performance still be considered stable or robust?**
>
> Yes, a model can have high task-specific performance but low RSS. This would however mean it is brittle and harder to deploy reliably. This underscores the value of explicitly measuring stability and scalability alongside performance.
>
> **4. Theoretical analysis for generalizing findings beyond the empirical settings.**
>
>  We appreciate the reviewer’s interest in generalizing the findings. While our focus is empirical, we hypothesize that a key factor driving behavior is the internal-external knowledge tradeoff: models with strong internal priors but weak integration of external input may degrade with increased retrieval.
>
> Our empirical results support this. For example, LLAMA begins with high no-context performance, and, at low k, its with-context answers closely mirror its no-context outputs—indicating strong reliance on internal knowledge. However, as more documents are retrieved, LLAMA’s answers diverge from its initial predictions (as shown in the attached figure), yet performance worsens. This suggests that LLAMA is incorporating context—but in a way that disrupts rather than improves its predictions. In contrast, FLAN shows lower no-context overlap from the start, adapts more readily to retrieved information, and maintains more stable performance as k increases.
> We will incorporate this framing into the revised draft to better contextualize model behavior across tasks and domains.
>
> **5. A model like FLAN may benefit from retrieval more consistently than GPT-3.5, but are these differences purely due to the models’ architectures, or are there other factors like training data or task-specific tuning influencing the outcomes?**
>
> Regarding FLAN-T5 being more scalable than GPT-3.5, our observations suggest that differences could arise from both architecture and training objectives:
>
> - FLANT5 explicitly has denoising training, where it learns to reconstruct original text from corrupted inputs, potentially helping it handle noisy or partially irrelevant retrieval contexts better. GPT-3.5, in contrast, does not explicitly train with a denoising objective, possibly explaining its comparatively limited scalability.
> - FLANT5 (encoder-decoder) explicitly encodes the retrieved context separately from decoding (generation), potentially enabling it to manage larger context sets more effectively. GPT-3.5, being decoder-only, processes context in a strictly autoregressive manner, which may cause diminishing returns as the amount of context increases due to difficulty in attending equally well to all retrieved information.
>
> **6. Related Work: REALM and FiD**
>
> Thank you for raising this. We agree that REALM [Guu et al., 2020] and FiD [Izacard & Grave, 2021] are foundational RAG systems:
> REALM introduced end-to-end retriever-reader pretraining and FiD showed the benefits of fusing multiple retrieved passages via decoder-only models.
>
> Our work is complementary: RAGGED offers tools to analyze reader-retriever behavior under varying retrieval depth and noise—dimensions not explicitly explored by these prior works. We will expand our Related Work section to reflect these connections.
>
> **7. Other Comments Or Suggestions** Thank you for the suggestions about the figures and formatting. We will fix them in the revision.

---

### Official Review · Reviewer_u17T · 2025-03-16

**Overall Recommendation:** 3

**Summary:**

This paper introduces RAGGED, a systematic framework for evaluating Retrieval-Augmented Generation (RAG) systems, focusing on stability, scalability, and robustness to noise. The authors analyze how retrieval depth, retriever-reader interactions, and dataset characteristics influence RAG performance, challenging the assumption that stronger retrievers universally improve results.

**Claims And Evidence:**

The majority of claims are well-supported by systematic experiments and cross-model/dataset validation. Key limitations (threshold justification, closed-model opacity) do not invalidate the core findings but highlight areas for future work. The evidence convincingly demonstrates that reader robustness—not retriever quality—is the critical factor in RAG performance.

**Essential References Not Discussed:**

Some related work about Adversarial Training of RAG.

**Experimental Designs Or Analyses:**

The experimental designs are largely sound for the paper’s goals, with rigorous testing across datasets, retrievers, and readers. However, the validity of conclusions is partially limited by arbitrary metric thresholds, insufficient statistical testing, and synthetic noise assumptions. Addressing these issues would strengthen the framework’s generalizability and robustness claims.

**Methods And Evaluation Criteria:**

The methods and evaluation criteria are well-suited for the paper’s goals. The framework’s design—diverse datasets, retrievers, readers, and noise conditions—provides actionable insights into RAG stability and scalability. While the novel metrics (RSS/RSC) and semantic validation could be refined, they represent a meaningful step toward standardized RAG evaluation. The experiments convincingly demonstrate that reader robustness, not retriever quality, is the critical factor, validating the utility of the framework.

**Other Comments Or Suggestions:**

1. Include adversarial perturbations (e.g., query paraphrasing, passage rewriting via LLMs) and temporal noise (e.g., outdated documents) to enhance robustness evaluation.

2. Benchmark against retrieval attacks to stress-test RSS.

3. Add a subsection analyzing the relationship between noise magnitude (e.g., % of corrupted passages) and RSS, possibly deriving error bounds.

**Other Strengths And Weaknesses:**

Strengths:

1. The introduction of RSS (Retrieval Scale Sensitivity) and RSC (Retrieval Scale Cost) metrics formalizes robustness and scalability evaluation for RAG systems in a unified manner, addressing a gap in prior work that often treated these aspects separately.

2.  Integrates concepts from adversarial robustness (e.g., noise injection) and computational efficiency (e.g., scaling with corpus size) into a cohesive evaluation paradigm, bridging domains like adversarial ML and distributed systems.

3. Provides actionable insights for practitioners to benchmark and optimize RAG systems, particularly in noisy or large-scale environments (e.g., enterprise search, customer support).

Weaknesses:

1. Relies on random non-relevant passages for robustness testing, neglecting adversarial or temporally inconsistent noise (e.g., adversarial retrieval attacks, outdated facts), which limits practical relevance.

2.  Experiments may not explore extreme-scale corpora (e.g., billions of documents), leaving scalability claims incomplete.

3. Absent theoretical guarantees (e.g., bounds on RSS/RSC under noise, convergence properties), leaving the framework purely empirical.

**Questions For Authors:**

1. Your robustness experiments inject random non-relevant passages as noise. Could your conclusions about RSS hold under more realistic noise scenarios, such as adversarial perturbations (e.g., passages with semantically similar but incorrect answers) or outdated documents? If not, how might this limitation affect the practical applicability of RSS?

2. The scalability analysis assumes a fixed retrieval depth $k$. How would RSC metrics change if evaluated with adaptive retrieval strategies (e.g., FLARE’s iterative retrieval based on uncertainty)? Could such strategies invalidate the trade-offs observed in your experiments?

3. Can you provide formal analysis (e.g., bounds) linking noise magnitude (e.g., % of corrupted passages) to RSS scores? For example, is there a threshold beyond which performance degradation becomes inevitable, regardless of retriever architecture?

**Relation To Broader Scientific Literature:**

The RAGGED framework advances the literature by formalizing robustness and scalability metrics for RAG systems, contextualizing instruction tuning’s role in noise tolerance, and validating semantic evaluation with LLM judges. Its innovations build directly on foundational RAG, robustness, and scaling literature while addressing underexplored challenges in real-world deployment. However, deeper engagement with fine-tuning-based RAG methods and realistic noise models would strengthen its positioning.

**Theoretical Claims:**

No theoretical claims are proposed in this paper.

---

> ### Author Rebuttal · Authors · 2025-04-01
>
> We sincerely thank the reviewer for the thoughtful and constructive feedback. We especially appreciate the recognition of how RAGGED addresses a gap in the literature by providing a unified framework for systematically evaluating scalability and stability and deriving actionable insights for real-world deployment.
>
> **1. Noise Assumptions in Robustness Evaluation**
>
> We appreciate the reviewer’s clarifying question about the robustness of our setup. To clarify: our “noise” is not artificially injected or random. Rather, it consists of non-gold passages retrieved by a real retriever. These are often topically relevant but incorrect, representing a realistic and common failure mode in deployed RAG systems. This setup reflects naturally occurring retrieval imperfections that practitioners routinely encounter.
>
> We agree that adversarial or temporally inconsistent noise (e.g., contradictory or outdated passages) is an important direction for future robustness research. However, modeling those scenarios requires assumptions (e.g., degree of contradiction, factuality, intent) that are beyond the scope of this work. That said, RSS could readily be extended to those settings, and we hope future work on adversarial or knowledge-conflict RAG systems will incorporate it.
>
> **2. Applicability to Extreme-Scale Corpora**
>
> We agree that testing scalability on corpora with billions of documents would be highly valuable. However, such datasets with annotated gold passages are currently limited. RAGGED is designed to be scalable by construction and can be readily applied to larger corpora as they become available.
>
> **3. Empirical vs. Theoretical Foundations**
>
> We appreciate the reviewer’s point. Our focus is empirical by design, aiming to provide actionable metrics for analyzing RAG system behavior in real-world settings. This mirrors the trajectory of many standard metrics (e.g., BLEU, ROUGE, F1), which were adopted for their practical value before receiving formal analysis. That said, we agree that theoretical grounding—such as bounding the relationship between noise and RSS—would strengthen the framework, and we will note this as an important direction for future work.
>
>
> **4. How would RSC metrics change if evaluated with adaptive retrieval strategies (e.g., FLARE’s iterative retrieval based on uncertainty)? Could such strategies invalidate the trade-offs observed in your experiments?**
>
> This is an excellent and insightful point. Our current analysis focuses on the standard retrieve-then-generate paradigm with a fixed top-k retrieval depth, which remains a common baseline in RAG systems. However, the core idea behind RSC—measuring how performance scales with increased retrieval—can be naturally extended to adaptive retrieval strategies like FLARE.
> In fixed-depth retrieval, RSC reflects the trade-off between performance gains and increasing the retrieval cutoff. In adaptive systems like FLARE, a similar trade-off exists between performance and the number or frequency of retrieval calls as determined by an uncertainty threshold. One could analogously define an RSC-style metric that varies the retrieval triggering threshold and measures how performance scales with the total retrieval budget. This would preserve the core RSC insight: quantifying trade-offs between retrieval cost and performance gain. We will mention this extension as a valuable future direction.
>
> **5. Justification of Metric Thresholds**
>
> We chose ε = 0.5 for RSC based on empirical analysis: the standard deviation of F1 across all models and k values is consistently below 0.5 (mean ≈ 0.38, max = 0.46). This makes 0.5 a conservative but meaningful threshold that exceeds metric noise. Moreover, increasing ε to 0.6 or 0.7 preserves model ranking, indicating robustness to the threshold choice.
> For RSS, we use a window of δ = ±5, which aligns with practical tuning ranges and matches how performance curves behave in most models (either plateauing or gently peaking). We also tested δ = 10 and found rankings unchanged. Smaller windows (e.g., δ = 1) lead to RSS variances across models of < 0.002, making the metric insensitive and less useful.
>
> **6. Statistical Robustness and Variability**
>
> To address concerns about statistical rigor, we analyzed standard deviations across k values for each model and confirmed that they remain consistently low (all < 0.5). Additionally, we verified that model rankings remain stable under variations in ε and δ. These findings support the reliability and robustness of our conclusions, and we will include supporting variance analyses in the revised draft.

---

### Decision · Program_Chairs · 2025-05-01

**Decision:**

Accept (poster)

**Comment:**

This work addresses the problem of systematically evaluating RAG systems. It argues that RAG systems are sensitive to system configuration and their effectiveness varies with the configuration, often resulting in suboptimal or degraded performance. It proposes a framework, RAGGED, as a step towards configuring RAG systems in an informed manner. The framework is intended to enable RAG engineers to evaluate their systems across diverse retriever-reader configurations, retrieval depths, and datasets. The work applies the proposed framework  open-domain, multi-hop, and specialized-domain datasets and the key insight obtained from the analysis is that how well the reader model deals with noise has the most significant influence on the effectiveness of RAG systems. The work proposes two metrics to quantify the robustness of RAG systems, RSS and RSC. RSS is intended to measure the consistency of the system's performance around its optimal retrieval depth. RSC is intended to measure the accumulated performance gain over the largest window where the model continues to gain from the increased retrieval depth.

The empirical study is done on three retrievers one from each type: BM25 representing classical IR system, ColBERT representing neural retrievers and Contriver representing dense retrievers. Several open source and closed source reader models are evaluated. Three QA datasets are used for evaluating RAG performance in the study with one representative from each of general knowledge, complex reasoning and domain-specific retrieval. To evaluate retriever performance recall@k and to evaluate reader performance unigram F1 measures are used.

The empirical study asks several questions and attempts to find answers for them. The first question is on the conditions that enable superior RAG performance over closed-book generation. A sub-question asked is does retrieval always help. The answer is no as some models benefit from retrieval whereas others don't. It is suggested that retrieval is not inherently beneficial and it is hypothesized that handling noise in the retrieval results is crucial to get consistent benefits. Another sub-question asked is does retrieval help all tasks and domains uniformly. The answer is again no as it is found that some tasks benefit more than others (ulti-hop questions vs single-hop questions).

The second major question is on the impact of retrieval depth on stability and scalability. The study finds that in some systems, improve-and-plateau trend is seen and in others peak-and-decline trend is observed. In summary, increasing retrieval depth is not always beneficial.

The third major question is on how noise affects the performance of the reader. To answer the question, a comparison is made between two scenarios - first scenario where at least one gold passage is present and the second scenario where no gold passage is present. The key finding is that most models perform worse than no-context baseline when no gold passages are retrieved. Further, the study finds that prompting with explicit relevance instructions improve noise filtering doesn't consistently help across the models. Based on this observation, the study recommends fine-tuning readers for noise resilience over retrieval-side interventions.

The study also investigates the influence of retriever choice on RAG system performance. It finds that retriever quality improves recall but doesn't necessarily improve the reader's performance. Similarly, it finds that reranking doesn't always improve the reader's performance.

The study recommends tuning of retrieval depth for every model and  fine-tuning readers for stability.

Reviewers agree that the claims made in the submission are backed by systematic experimentation and that the study provides potentially useful insights to practitioners. Reviewers expressed concern about the generalizability of the findings as a) it is not backed by theoretical analysis b) retrieval systems and models considered in the study are not state-of-the-art c) benchmarks used for evaluation are dated. Reviewers also expressed concern about a) the soundness of the proposed measures for stability and scalability b) focusing too narrowly on stability of RAG systems without giving task performance its due importance. Reviewers made some good suggestions to strengthen the work further by a) benchmarking against retrieval attacks to stress-test the proposed measures b) evaluating systems with adaptive retrieval strategies c) including SOTA models and newer benchmarks in the study.

Despite its aforementioned limitations, the work makes a strong case for rethinking the evaluation of RAG systems. As one of the reviewers put it, the work "provides a more refined view of RAG, emphasizing the importance of reader robustness and domain-aware retrieval strategies, while challenging the oversimplified notion that stronger retrieval always leads to better performance. These contributions push the research further in understanding and optimizing the interaction between retrievers and readers for RAG systems."